# Central Sensitisation and functioning in patients with chronic low back pain: protocol for a cross-sectional and cohort study

Jone Ansuategui Echeita ![ORCID],[1] Henrica R Schiphorst Preuper,[1] Rienk Dekker,[1] Ilse Stuive,[1] Hans Timmerman,[2,3] Andre P Wolff,[3] Michiel F Reneman[1]

[1]Department of Rehabilitation Medicine, University of Groningen, University Medical Center Groningen, Groningen, The Netherlands
[2]Department of Anesthesiology, Pain and Palliative Medicine, Radboud University Medical Center, Nijmegen, The Netherlands
[3]Department of Anesthesiology Pain Center, University of Groningen, University Medical Center Groningen, Groningen, The Netherlands

**Correspondence to**
Ms Jone Ansuategui Echeita;
j.ansuategui.echeita@umcg.nl

## ABSTRACT

**Introduction** A relevant subsample of patients with chronic low back pain (CLBP) have manifested augmented central pain processing, central sensitisation (CS). Patients with CLBP have limited functioning and participation. Theoretically, physical functioning in patients with CLBP can plausibly be linked to CS; however, evidence to explain such association is scarce. Moreover, there is no gold standard for CS diagnosis. The objectives of the study are: (1) to analyse the association between instruments assessing reference symptoms and signs attributed to CS; (2) to analyse whether reference symptoms and signs attributed to CS are associated with functioning measurement outcomes; and (3) to analyse whether changes (between baseline and discharge) in reference symptoms and signs attributed to CS are related to changes in each of the functioning measurement outcomes.

**Methods and analysis** A cross-sectional and longitudinal observational study is performed with measurements taken at baseline and discharge of an interdisciplinary rehabilitation programme. A sample size of 110 adult patients with CLBP has been calculated for the study. CS measurements are: Central Sensitisation Inventory, quantitative sensory testing and heart rate variability. Functioning measurements are: lifting capacity, maximal aerobic capacity, accelerometry and reported functioning. Statistical analyses to be performed are: (1) correlation between CS measurements, (2) multiple regression between functioning (dependent variable) and CS measurements (independent variable), and (3) multiple regression between changes in scores of functioning (dependent variable) and CS measurements (independent variable), and corrected for sex and age.

**Ethics and dissemination** The study obtained the clearance to its implementation from the Medical Research Ethics Committee of the University Medical Center Groningen in July 2017. The results will be disseminated through scientific publications in peer-reviewed journals, presentations at relevant conferences, and reports to stakeholders.

**Trial registration number** NTR7167/NL6980.

## INTRODUCTION
### Background
Twenty per cent of the general population experiences prolonged or recurrent pain for

### Strengths and limitations of this study

► Novel and comprehensive approaches for the assessment of the presence of central sensitisation (CS).
► Extensive amount of measurements from multiple domains for the determination of functioning status.
► Longitudinal study with measurements at baseline and discharge of an interdisciplinary rehabilitation programme implemented as 'Care As Usual'.
► Strategies allocated to promote participation and adherence, and to mitigate selection bias to the largest extent possible.
► Statistical analyses are performed with selections of CS and functioning measurements, and these selections may influence study results.

at least 3 months, such pain is then considered to be chronic.[1 2] Low back pain is the principal cause of years lived with disability and workdays lost in Europe and worldwide,[3] and its sequelae affect 9% of the global population.[4] Moreover, although there are several treatment intervention strategies available for patients with chronic low back pain (CLBP), treatment effect sizes have only been moderate.[5] Such evidence may explain why 40% of the patients are not satisfied with the treatment received.[2] The economic consequences of chronic pain include direct costs of medical care and pain management treatments, as well as indirect costs of decreased work productivity and sick leave, and of disability and sickness retirement.[2 6] The indirect costs determine 86%–88% of the total costs produced by patients with CLBP.[7 8] In particular, in the Netherlands, the costs derived from back pain to Dutch society were 3.5 billion in 2007, corresponding to 0.6% of the gross national product.[8]

In the past decennia, a relevant subsample of patients with chronic pain, including individuals with CLBP, have shown an increased responsiveness to noxious and nonnoxious stimuli, described as central sensitisation (CS).[9] CS is a result of an imbalance in the nociceptive pathways ('pain pathways') and supraspinal structures due to an amplified facilitation and/or reduced inhibition.[10–12] The phenomenon, CS, is manifested by an amplified pain perception regarding its intensity (hyperalgesia and allodynia), duration (aftersensations and temporal summation) and distribution (expansion of the receptive field), as well as a reduced conditioned pain modulation (CPM).[12] There is currently no gold standard for CS diagnosis. Identifying reference clinical symptoms and signs, however, may indicate the presence of CS: disproportionate pain, diffuse distribution, allodynia and hyperalgesia, hypersensitivity unrelated to mechanical stimulus but rather to environmental sensations (light, temperature, noise or other stressors), maladaptive psychosocial factors and low vagal nerve activity.[13–16]

Chronic pain is a multifactorial condition which has an impact on physical or body structures and functions, on psychological processes and on daily activities and quality of life of individuals.[2 17 18] Limited functioning or capacity to perform everyday tasks (essential for an independent living) affects individuals with chronic pain in their mobility and self-care, social relationships, work and leisure.[19] Individuals' functioning is composed of body functions (body level)—described by impairments they experience—and of activities and participation (personal and social levels, respectively)—described by capacity and performance tests.[18 20] An extensive overview of the functioning status is best obtained when the assessment includes suitable and relevant measures from multiple domains of daily living. In particular, lifting capacity, physical activity (PA), aerobic capacity or daily activities participation are measures that account for the multiple operationalisations of functioning in the daily living.

Patients with CLBP are likely to be less physically active, and be more deconditioned and disabled.[3 21–23] A subsample of patients with CLBP can manifest symptoms and signs of CS being present.[9] Because patients with CS are more reactive to stimuli and their pain experience is enhanced, the presence of CS and/or higher levels of CS could plausibly be associated with lower physical functioning and participation in patients with CLBP, and vice versa. More severe symptomatology of CS, decreased parasympathetic/vagal activity or altered somatosensory responses have been associated with greater self-reported pain-related disability.[24–30] However, the associations between altered somatosensory responses and self-reported disability were dependent on the site of testing, the stimuli and the type of tests performed. Also, CS has been assessed with different instruments that measure specific CS clinical symptoms,[12] lacking a comprehensive assessment of CS in which various reference clinical symptoms are measured together. Moreover, functioning has most frequently been assessed with self-reported disability

which is not exclusively representative of the functioning status of the patient.

The study described in this research protocol has been created to improve the assessment and management of pain and disability of centrally sensitised patients with CLBP and fill the existing gaps in knowledge. On the one hand, it intends to assist on the phenotyping of centrally sensitised patients with CLBP. To this purpose, an integrated assessment of CS combining the use of various instruments which measure different reference clinical symptoms of CS is aimed. Instruments like Central Sensitisation Inventory (CSI),[31] quantitative sensory testing (QST)[32 33] and heart rate variability (HRV)[34] are noninvasive, feasible and assess different reference clinical symptoms and signs indicative of CS.[12] In the absence of a gold standard, these instruments measure different reference indicators of CS and can also be complementary to describe CS. On the other hand, it intends to improve the understanding of CS phenomenon and its contribution to functioning status. Lifting capacity, PA levels and/or PA distribution or patterns, aerobic capacity and daily activities participation are representative of several tasks of the daily living usually limited in patients with CLBP. These capacity and performance tests and self-reported functioning measures can give an extensive overview of the functioning status.

## Objective(s)

The objectives of the studies covered by this research protocol are:

1. To analyse the association between three instruments assessing reference symptoms and signs attributed to CS in patients with CLBP: CSI, QST and HRV.
2. To analyse whether reference symptoms and signs attributed to CS are associated with functioning measurement outcomes in patients with CLBP. Specifically, to analyse the association between the three instruments assessing CS (CSI, QST and HRV) and: lifting capacity, aerobic capacity, PA levels and/or PA distribution or patterns, and patient-reported measurements.
3. To analyse whether changes (between baseline and discharge) in reference symptoms and signs attributed to CS are related to changes in each of the functioning measurement outcomes in patients with CLBP. Specifically, to analyse whether decreases in CS described as: a lower CSI score, an increase in the pain thresholds (QST) and an increase in parasympathetic/vagal tone (HRV) are related to a better functioning described as: an increase in lifting capacity, an increase in aerobic capacity, an increase in PA levels and/or a change in PA distribution or patterns, and better outcome scores in patient-reported measurements.

## METHODS AND ANALYSES
### Study design

An observational study project with both cross-sectional and longitudinal designs is being conducted from

September 2017 to September 2019 in the Center for Rehabilitation of the University Medical Center Groningen (CvR-UMCG) in the Netherlands.

## Patient and public involvement

The personnel of the CvR-UMCG collaborate in the study, in particular, the Pain Rehabilitation team during patient management and data collection process and the central exercise lab team on data collection process. The authors wish to express their gratitude to the collaborators and to the patients who participate.

## Participants

Consecutive patients between 18 and 65 years of age at the time of recruitment who are primarily referred to the outpatient Pain Rehabilitation Department of the CvR-UMCG due to a chronic primary low back pain and who are assessed for the interdisciplinary pain rehabilitation programme are eligible to participate in the study. Chronic primary low back pain is defined as a low back pain recurrent for more than 3 months, related to emotional distress and/or functional disability and whose pain is not a result of any other diagnosis[35] even if patients may have undergone surgery in the past. Additionally, patients are invited to the study if they are mentally competent and can follow instructions. Furthermore, patients are excluded from the study if, based on their file, they: have a specific diagnosis that would better account for the symptoms (cancer, osteoporosis, rheumatological inflammatory diseases and/or spinal fractures), have neuralgia and/or radicular pain in the legs, have severe psychiatric disorders, show sensitivity alterations in measurement location(s), are pregnant (or planning to be), have any specific contraindication according to the standards of practice guidelines of the assessments that they are going to be subjected to,[36 37] or take any medication that may interfere with the assessment outcomes such as calcium channel blockers on HRV. CS-related comorbidities (ie, fibromyalgia, osteoarthritis or chronic fatigue syndrome) are no reason for exclusion from the study. The use of opioid or pain-related medication is neither a criterion for exclusion and such information is recorded.

## Procedures

Measurements are performed as part of 'Care As Usual' (CAU) for outpatients following the pain rehabilitation guidelines of the CvR-UMCG. However, not all measurements of the research protocol belong to the battery of assessments of the CvR-UMCG, as pointed out in the legend of figure 1.

During the first appointment of the patients with the rehabilitation physician, patients are invited to participate in the studies if based on their file they are eligible according to the criteria defined for the studies. Eligible patients who are invited to participate and receive information about the study, but have not reacted, are contacted in a week's time to remind them about the study participation. Those who sign the informed consent, fill in forms with their relevant personal information during the baseline assessment and are given vouchers for the travel expenses during the discharge assessment. Participating patients follow CAU trajectory with the Pain Rehabilitation team in terms of assessments, screening and implementation of the interdisciplinary rehabilitation programme. As part of the research protocol, the performance of additional assessments and filling up of questionnaires on CS and functioning are required. CSI, lifting capacity and HRV results are collected as CAU. Aerobic capacity, QST, objective PA monitoring, and pain intensity and functioning questionnaires' results are collected as additional to CAU. All CS measurements, objective PA monitoring and capacity tests are measured at two time points, baseline and discharge. Pain intensity and functioning questionnaires, on the other hand, are measured at 13 time points, baseline and once a week during the 12-week rehabilitation programme. Pain Disability Index (PDI) is the only functioning questionnaire collected as CAU at baseline but as additional during the rehabilitation programme. A diagram of the project procedure is presented in figure 1.

## Measurements

### CS measurements

In order to comprehensively assess the symptoms and signs attributed to CS, data are obtained from a questionnaire (CSI) and two physical assessments (QST and HRV).

### *Central Sensitisation Inventory*

The inventory is a self-reported questionnaire with two sections to estimate the severity of CS-related symptoms. Section A is a 25-item questionnaire to assess the presence of the most common CS-related symptoms. Each of the questions is measured on a 5-point Likert scale from 0 (never) to 4 (always), and the total score ranges from 0 to 100, where higher scoring is associated with more severe symptomatology. Section B is a 10-item questionnaire to assess the previously diagnosed CS syndromes (CSS) and/or conditions related to CS. Higher amount of CSS is associated with higher chances of CS being present. This instrument has shown excellent test–retest reliability (intraclass correlation coefficient (ICC)=0.82–0.97) and internal consistency (Cronbach's α=0.87–0.91) in the general population and in patients with chronic pain,[31 38–40] as well as good concurrent validity (sensitivity=81%, specificity=75%) for CSS diagnosis within patients with chronic pain.[38] For the purposes of the study the Dutch translation of the CSI is used,[41] which has also shown excellent internal consistency (Cronbach's α=0.91) and test–retest reliability (ICC=0.91 and 0.88) in controls and in patients with chronic musculoskeletal pain, respectively.[42] The psychometric information of CSI in patients with CLBP is not known.

### *Quantitative sensory testing*

A battery of standardised sensory tests quantifies the function of the somatosensory system. QST tests the

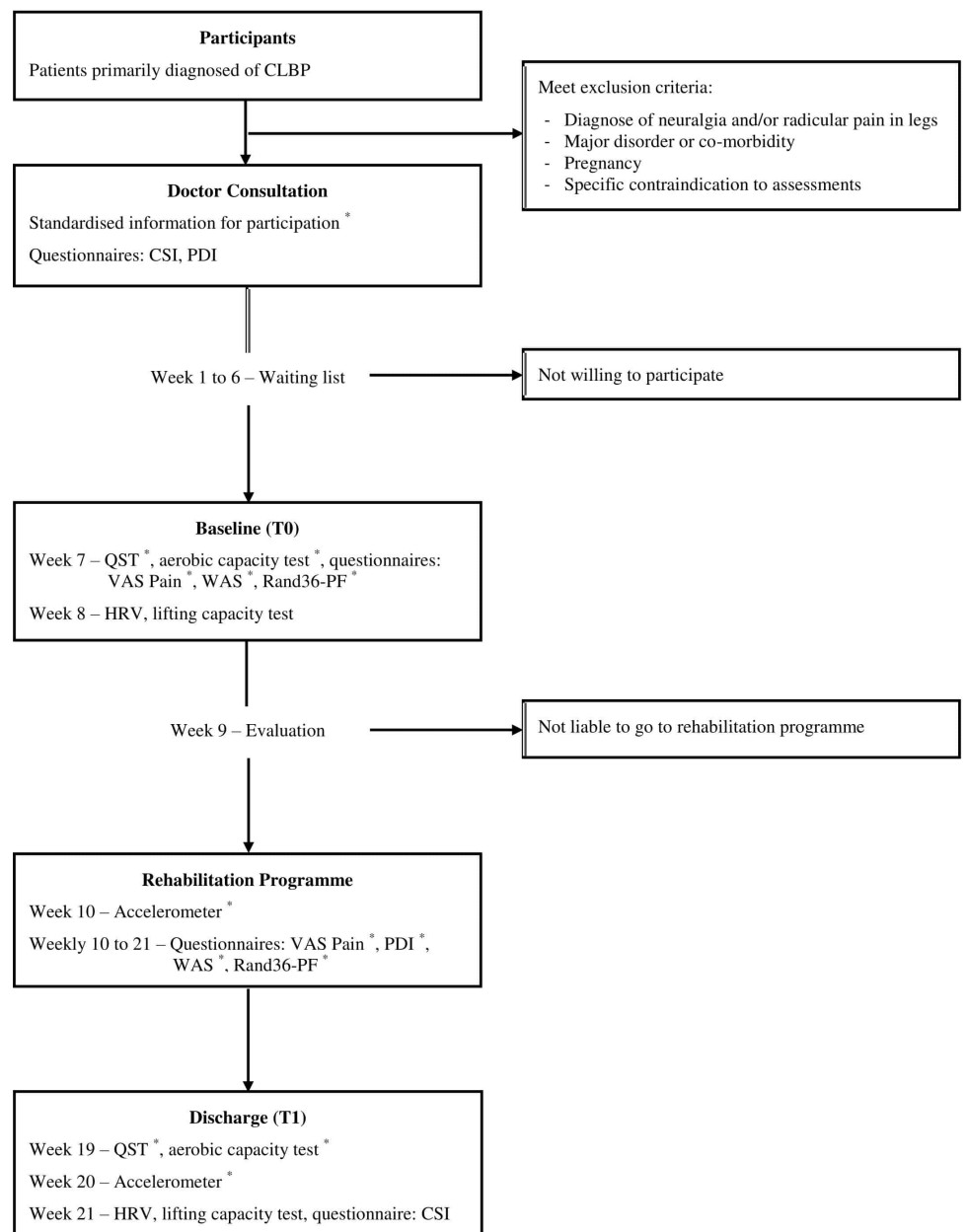

**Figure 1** Flow chart of the project procedure. *Measurements obtained which do not belong to Care As Usual (CAU) of the Pain Rehabilitation Department of the Center for Rehabilitation of the University Medical Center Groningen (CvR–UMCG). CLBP, chronic low back pain; CSI, Central Sensitisation Inventory; HRV, heart rate variability; PDI, Pain Disability Index; QST, quantitative sensory testing; Rand36-PF, Rand36 Physical Functioning subscale; VAS, Visual Analogue Scale; WAS, Work Ability Score.

characteristics of various sensation and nociceptive processes as representative of afferent nerve fibres and central pathways performance. The whole assessment lasts for 1 hour, and it consists of six sensory tests, which are measured in seven different predefined body locations: one training location to familiarise the patient with the tests, the most painful location (as pointed out by the patient), and five control locations. See online supplementary material 1 for full description of body locations. The standardised instructions of the sensory tests are read to the patient at each test performance. Of the tests, five consist of mechanical stimuli based on and performed

following the QST examination of the German Research Network on Neuropathic Pain (DFNS) protocol,[32] and one for CPM following the Nijmegen-Aalborg Screening QST protocol.[33 43] Overall, the DFNS protocol has shown good test–retest (mean r=0.86) and interobserver reliability (mean r=0.83) in patients with affections in the somatosensory system either with or without pain.[44] For the purpose of avoiding bias due to the order of the measurements and locations, a counterbalancing procedure is established, which accounts for the locations and the measurements to be made. This procedure provides three different assessment order groups, all of which

begin with the measurements at the training location and end with the CPM test. An additional counterbalancing procedure has been made for allocating patients to these assessment order groups (either 1, 2, or 3). Every nine patients, a different allocation order is performed to ensure a similar amount of QST assessments per group. All assessors are trained in the QST protocol. The following sensory tests are performed:

► *Dynamic mechanical allodynia* (*DMA*)—to test the presence of abnormal stimuli-evoked pain—is assessed with a soft brush (SENSElab, Brush-05, Somedic, Hörby, Sweden) that is swiped 1–2 cm over the skin. This is performed on a single repetition and on a train of three repetitions (wind-up response). At each of the seven body locations (see online supplementary material 1) it is recorded whether the patients feel the sweeping and its pain intensity with a numeric rating scale (NRS) ranging from 0 (no pain) to 100 (most intense pain imaginable) for both the single swipe and the series.[45 46] The DMA (ratio) is calculated as: DMA=(NRS-3/NRS-1). If NRS-1 is 0, the DMA for that location cannot be calculated and it will be registered as a missing value. The DMA test according to the DFNS protocol has shown to have good test–retest (r=0.87) and interobserver reliability (r=0.79).[44]

► *Mechanical pain threshold* (*MPT*)—to test the ability to perceive sharp, pricking or stinging stimuli—is assessed with a set of standardised weighted pinprick stimulators with seven different intensities (the PinPrick Stimulator Set, MRC Systems, Germany) and with a modified method of limits. Starting at the lowest intensity (8 mN), it progressively increases until patients report the stimulus to change from touch to painful (sharp, pricking or stinging). The intensity of the pinprick that produces the first painful stimulus in millinewtons is recorded in each of the seven locations (see online supplementary material 1).[46–49] The MPT test according to the DFNS protocol has shown to have good test–retest (r=0.80) and interobserver reliability (r=0.80).[44]

► *Wind-up ratio* (*WUR*)—to test the nervous system's painful stimuli temporal summation or responsiveness to repeated noxious stimuli. Such test is assessed with a suprathreshold pinprick stimulator of 256 mN (the PinPrick Stimulator Set, MRC Systems, Germany), which is applied on a single repetition and on a train of 10 repetitions in an area of 1 cm$^2$ with intervals of 1 s (wind-up response). The pain intensity measured with an NRS ranging from 0 (no pain) to 100 (most intense pain imaginable) for both the single repetition and the series is recorded in each of the seven body locations (see online supplementary material 1).[50 51] The WUR (ratio) is calculated as: WUR=(NRS-10/NRS-1). If NRS-1 is 0, the WUR for that location cannot be calculated and it will be registered as a missing value. The WUR test according to the DFNS protocol has shown to have poor test–retest (r=0.67) and interobserver reliability (r=0.56).[44]

► *Mechanical detection threshold* (*MDT*)—to test the ability to detect light touch—is assessed with a standardised set of von Frey filaments (OptiHair2-Set, Marstock Nervtest, Germany) with a modified method of limits.[52 53] Starting at the intensity of 16 mN, the stimulus application progressively decreases until patients do not report any sensation. The intensity of the filament that is not perceived in millinewtons is recorded in each of the seven body locations (see online supplementary material 1).[54] The MDT test, according to the DFNS protocol, has shown to have good test–retest (r=0.90) and interobserver reliability (r=0.89).[44]

► *Pressure pain threshold* (*PPT*)—to test the ability to perceive pressure pain stimuli on muscles—is assessed with an electronic pressure algometer (Force Ten FDX, Wagner Instruments, USA). With this tool, the pressure in the contact point (1 cm$^2$ tip) is progressively but steadily increased until patients report to feel pressure change towards painful (burning, drilling or aching). The intensity in newtons when pressure becomes painful and its pain intensity with an NRS ranging from 0 (no pain) to 100 (most intense pain imaginable) scale is recorded in each of the seven body locations (see online supplementary material 1).[32] The PPT test according to the DFNS protocol has shown to have good test–retest (r=0.88) and interobserver reliability (r=0.84).[44]

► *CPM*—to test the nervous system's descending pain processing. The CPM test methodology selected for this study follows the recommendations of the EFIC meeting (2013).[55] The pain modulation is assessed by means of PPT tests (test stimuli) in a distant location from patients' dominant hand (musculus deltoid and musculus quadriceps of the non-dominant side, see online supplementary material 1) before and after immersing the dominant hand in a bucket full with ice and cold water (cold conditioning stimuli). First, PPT measurements at distant locations are made; then, patients immerse the dominant hand in the bucket for as long as they possibly can with a maximal duration of 3 min; finally, PPT measurements are repeated. The PPT results, intensity in newtons and pain intensity with an NRS ranging from 0 (no pain) to 100 (most intense pain imaginable) scale, in both locations and the time the hand is immersed are recorded. The CPM (%) is calculated as: CPM=[(PPTpost–PPTpre)/PPTpre]∗100, where lower values indicate lower alteration on the nociceptive pathways.[33 43] The CPM test, as the combination of the PPT handheld algometer and cold stimuli, has shown to be one of the most reliable methods with a modest test–retest reliability (ICC=0.49 and coefficient of variation=63.6%).[56]

### Heart rate variability
The test provides an indication of the function and balance of the autonomic nervous system (ANS), which is responsible for the control of internal functions including self-regulation, adaptability and resilience

capacities.[57] The ANS is susceptible to mental and emotional processes, which in turn can produce an imbalance in self-regulation (affecting, among others, heart rate (HR)) by a hyperactivation of the sympathetic tone and reduced parasympathetic/vagal tone.[15 34] HRV testing can monitor heart rhythms, or time and pattern variation on HR and interbeat (RR) intervals, and can be used to assess the function of the ANS.[58 59] The HRV test is performed by assessors trained by the HeartMath Institute, and the 5 minute protocol is executed to guarantee the comparison of the results as recommended by the task force for short-term recordings.[58] Patients are instructed to sit calmly and breathe normally, while they have an ear pulse sensor placed which is connected to a computer.[60] The following measures are collected and recorded with the emWave PC software (emWave, HeartMath, Boulder Creek, California, USA): mean HR in beats per minute, mean RR interval in milliseconds; time domain measures in milliseconds as SD of normal-to-normal interval indicator for cyclic components responsible of RR interval variability, root mean square of successive differences (RMSSD) for beat-to-beat variance in HR and parasympathetic/vagal tone; frequency domain measures in $ms^2/Hz$ as very-low-frequency (0.0033–0.04 Hz) indicator for long-term regulation and sympathetic tone, low frequency (0.04–0.15 Hz) for sympathetic and parasympathetic/vagal tones and high frequency (0.15–0.40 Hz) for parasympathetic/vagal tone related to breathing rate; and normalised coherence indicator for HRV stable or regular pattern over time in percent, where higher scores mean more harmonic signal.

### Physical functioning measurements

In order to determine the functioning status of individuals with chronic pain in multiple domains, data are obtained from two physical assessments (lifting and aerobic capacity tests), an accelerometer (PA level) and a battery of reported functioning questionnaires.

### *Lifting capacity test*

The floor-to-waist lift capacity test is based on the Work-Well Functional Capacity Evaluation (FCE) protocol[61] and is performed to measure patients' work functioning or capacity. Such FCE lift test has been shown to have high test–retest (ICC=0.81) and inter-rater reliability (ICC=0.76) in patients with CLBP.[62 63] Assessors are trained in the floor-to-waist lift capacity test procedure. Patients are given standardised instructions to repetitively lift a crate with weights from a shelf at waist height to the floor and back. The test begins with a weight that can easily be lifted, progressively the load is increased and the test ends when the maximum capacity is reached. The final endpoint can be reached due to: cardiac endpoint (85% of maximum HR), biomechanical endpoint (unsafe increasing weight because of lack of load handling control), patient endpoint (patient decided to stop) and criterion endpoint (normal end of the test).[64] The maximal load lifted in kilograms is recorded along with

the endpoint reason, and the patient-reported and the assessor-observed mechanical effort as measured with Borg's category ratio (CR-10) scale.[65] Borg's CR-10 scale ranges from 0 (at rest) to 10 (very, very hard), and has demonstrated to be a valid visual observation of the effort level of patients during their performance of a lifting capacity test.[66]

### *Aerobic capacity test*

The maximal aerobic capacity is assessed with a cardiopulmonary exercise test (CPET). The test is performed with a cycle ergometer (Ergoselect 200p or Ergoselect 200k, Ergoline, Bitz, Germany) following a defined continuous ramp protocol. Before any cycling begins, patients are interviewed about their daily activities, and their resting heart rhythm and function are recorded with an ECG (Assy Cam-14, GE Medical Systems, USA). Patients then have an unloaded warm-up of 3 minutes at 60–70 rotations per minute before the test begins. At the start of the test, an experienced exercise physiologist in discussion with a specialised physician or nurse determines the starting workload (25–100 W) and ramp (5–25 W increase per minute) depending on patients' estimated fitness level. During the test, patients are asked to maintain a constant cadence while the working load is progressively being increased. Also, they are continuously being verbally encouraged to obtain their maximum performance. The maximal performance is determined by various parameters: a temporary loss of strength and energy (=exhaustion), a plateau in the peak oxygen uptake ($VO_2$max), a respiratory exchange ratio higher than 1.15 and/or an HR higher than 85% of the predicted maximal HR.[37] Overall, the test lasts for 8–12 minutes and throughout the whole assessment patients are monitored on their cardiac activity with an ECG, on their blood pressure with a cuff algometer and on their ventilatory gases on a breath-by-breath basis with a metabolic cart (JAEGER Vyntus CPX, Jaeger, Germany, Hoechberg). The $VO_2$max in mL/min and per kilogram in mL/min/kg is collected as the mean of $VO_2$ over the last 30 seconds of exercise. Along with $VO_2$max, the following measures are collected: peak workload in watts, HR at end of test in beats per minute, breathing frequency at end of test in breaths per minute, expiratory tidal volume in litres, peak energy expenditure in metabolic equivalent of task (MET), test duration in minutes, and the patient-reported and the assessor-observed mechanical effort as measured with Borg's rating of perceived exertion (RPE) scale. Borg's RPE scale ranges from 6 (no exertion at all) to 20 (maximal exertion)[65] and has shown to be highly correlated with HR (r=0.74) and blood lactate (r=0.83),[67] demonstrating to be a reliable instrument to measure work intensity.

### *PA levels*

The objective measurements of PA levels are obtained by means of an accelerometer (ActiGraph GT9X Link, ActiGraph, Pensacola, FL, USA). The accelerometer is a small and light device that monitors wearers' daily

PA and energy expenditure. Participating patients wear the accelerometer attached at their hip level for 1 week continuously during walking hours, in order to ensure the minimum of 4 full-day recordings as recommended by activity monitoring reviewers and experts.[68 69] Different algorithms are chosen from the software provided by the manufacturer, ActiLife6 (ActiGraph, Pensacola, Florida, USA), to discriminate between low PAs and split patients in three activity intensity categories (sedentary, light and moderate)[70] and to obtain patients' energy expenditure in kilocalories[71] and MET.[72] The daily and weekly time in hours and percentage of time spent in each of the activity intensity categories as well as the estimated energy expenditure in both kilocalories and MET are collected.

### Reported functioning

Data from this domain are obtained from four questionnaires sent to patients and filled up by them via internet. All the questionnaires require 10–15 minutes to be filled in.

▶ *Physical functioning* (*Rand36 Physical Functioning subscale—Rand36-PF*). The Rand36 is a health-related quality of life survey that measures the impact of health on patients' well-being and ability to function. Patients are asked to answer only the Physical Functioning (PF) subscale which looks at the limitations patients experience during their daily activities. Each of the 10 items belonging to the PF subscale has a three-level response choice ranging from 0 (limited a lot) to 100 (not limited at all), the total average score ranges from 0 to 100 too; where a higher score is associated with a greater limitation in performing daily activities. For the purposes of the study, the Dutch translation of the Rand36 is used, which has shown good internal consistency (Cronbach's α=0.92) and test–retest reliability (r=0.72–0.82) in the general population.[73 74]

▶ *Disability* (Pain Disability Index—*PDI*). The index is a 7-item questionnaire which measures the interference of pain with the ability to function in different daily life activities: family/home responsibilities, recreation, social activity, occupation, sexual behaviour, self-care and life support activity. Each item ranges from 0 (no interference) to 10 (total interference), and the total score ranges from 0 to 70; where scoring higher is associated with higher interference in daily life activities. The PDI has shown modest test–retest reliability (r=0.44) and high internal consistency (Cronbach's α=0.86) in patients with chronic pain.[75] For the purposes of the study the Dutch translation of the PDI is used, which has shown a good internal consistency (Cronbach's α=0.85) and test–retest reliability (ICC=0.76) in patients with musculoskeletal pain.[76]

▶ *Work ability* (*Work Ability Score—WAS*). The score is a single-item question which compares patient's current work ability with their lifetime best. The result is a score ranging from 0 (unable to work) to 10 (best working ability). The single-item question is part of the Work Ability Index (WAI), a questionnaire which measures work ability and has shown to be test–retest reliable (difference between test and retest=−0.53) in construction workers.[77] The WAS has shown to be highly correlated to the 28 items of the WAI among women on a long-term sick leave (r=0.87)[78] and among active workers (r=0.63).[79]

### Supplementary measurements
### *Demographic characteristics*

Patients fill in a form with their details when the QST assessment takes place. Such details include: age, sex, nationality, height, weight, pain characteristics (affected body area, duration and medication, if applicable), educational level and employment characteristics (work status and physical work demands per Dictionary of Occupational Titles).[80]

### *Pain-related questionnaires*

▶ *Pain intensity* (*Visual Analogue Scale—VAS Pain*). The scale is a single-item question to assess the current pain intensity in adults. It is a straight line with 'no pain' (score=0) at one end and 'pain as bad as it could be' (score=100) at the other, in which patients are asked to mark the line at the level that represents their pain level. The score is the distance between the mark and the 'no pain' end, where scoring higher is associated with greater pain intensity. The VAS Pain has shown good validity (r=0.62–0.91) and good test–retest reliability (literate r=0.94 and illiterate r=0.71) in patients with rheumatic pain conditions.[81] VAS Pain is sent to patients and filled in by them via internet along with the battery of functioning questionnaires (described above), that is, at baseline and weekly during the interdisciplinary rehabilitation programme.

▶ *Body response to a straining exercise* (*Pain Response Questionnaire—PRQ*).[82] The questionnaire assesses the pain intensity of the different body locations on an 11-point numeric scale (NRS) ranging from 0 (no pain) to 10 (worst pain imaginable), and the type of pain felt (muscle soreness, other, both or unknown). Additionally, it controls for the use of medication and for the performance of any other strenuous PA. For the purposes of this study, the body locations measured are predefined to four and the questionnaire is being filled in on three occasions: before the CPET, immediately afterwards and 24 hours later. PRQ is filled up at the same moment when the aerobic capacity test is performed (ie, during baseline and discharge assessments).

### *Psychological screening questionnaires*

These questionnaires measure psychosocial features common to CSS that have demonstrated to affect pain experience and functioning.[83] Patients fill in the questionnaires via internet that are sent per CAU procedures by the Pain Rehabilitation team. Such questionnaires include:

- *Pain Catastrophizing Scale (PCS)*. The 13-item questionnaire measures in a 5-point Likert scale (0—not at all, 4—all the time) the degree to which patients have had catastrophic thoughts or feelings during their pain experience and accounts for three subscales or dimensions: rumination, magnification and helplessness.[84] For the purposes of the study the Dutch translation of the PCS is used,[85] which has shown good internal consistency (Cronbach's α=0.85–0.91) and test–retest reliability (r=0.92) among healthy individuals (students) and patients with CLBP.[86]

- *Injustice Experience Questionnaire (IEQ)*. The 12-item reported questionnaire assesses in a 5-point Likert scale from (0—never, 4—all the time) four different aspects of the perceived injustice due to injury: severity, blame, unfairness and irreparability. The questionnaire has shown to have good internal consistency (Cronbach's α=0.92) and test–retest reliability (ICC=0.98) in a group of patients with fibromyalgia.[87] For the purposes of the study the Dutch translation of the IEQ is used, as translated by the Pain in Motion group.[88]

- *Brief Symptom Inventory (BSI)*. The inventory is a 53-item reported questionnaire which assesses patient's current level of psychological distress with 5-point Likert scale questions (0—not at all, 4—extremely), the amount of symptoms present and their severity.[89] BSI is composed of nine subscales: somatisation, obsession-compulsion, interpersonal sensitivity, depression, anxiety, hostility, phobic anxiety, paranoid ideation and psychoticism. For the purposes of the study the Dutch translation of the BSI is used, which has shown good internal consistency (Cronbach's α=0.71–0.87) and test–retest reliability (r=0.71–0.89) among psychiatric outpatient and healthy individual groups.[90]

## Sample size

Due to the limited amount of relevant previous or similar studies, the sample sizes for the several analyses described in this research protocol are estimated a priori with GPower (G*Power for Windows, Version 3.1.9.2), and are exploratory as a result.

The following sample size calculations, as well as the planned statistical analyses later described, will be performed with proxies for the main variables. Additionally, due to the influence of patient's sex and age in the assessment outcomes, all analyses will include these variables for a moderator effect.

- *Sample size for objective 1: to analyse the association between instruments assessing reference symptoms and signs attributed to CS*. Correlation analyses will be performed for this objective. The correlation between the different CS measurements is expected to be weak as reported on associations between certain CS measures and tests.[24] For a correlation coefficient of 0.25, an α error probability of 0.05 and a power of 0.8, a minimum sample size of 97 is calculated.

- *Sample size for objective 2: to analyse whether reference symptoms and signs attributed to CS are associated with functioning measurement outcomes; and for objective 3: to analyse whether changes (between baseline and discharge) in reference symptoms and signs attributed to CS are related to changes in each of the functioning measurement outcomes*. Multiple regression analyses will be performed for objectives 2 and 3. Based on the associations between CSI and psychosocial and cognitive behavioural factors,[24 25] the effect size is hypothesised to be medium. Since minimally three CS measurements are tested with sex and age required as independent variables, and controlled for the confounding effect of other demographic and test-related variables, eight predictors are foreseen. Taking into account the factors mentioned above, the sample size calculation results in a minimum of 77 patients, calculated with an α error probability of 0.05 and a power of 0.8. Finally, patients measured at discharge could be about 30% less than at baseline due to either not following the interdisciplinary rehabilitation programme or personal reasons; a sample size of 110 patients may be required in this scenario.

## Analyses

Data records from the different tests and questionnaires will be collected and merged into one database. The database management will be done by the project team and handled according to the internal regulations of the UMCG. The analyses of the study will be performed with SPSS software version 22.0 (IBM).

Before any analysis, data will be prepared:
- Variables need to be calculated: body mass index; DMA, WUR and CPM tests (see the Methods and analyses section, Measurements: CS measurements); PDI, Rand36-PF, PCS, BSI and IEQ questionnaires.
- HRV: data will be preprocessed to correct for any artefacts that may be present at the beginning or at the end of the recording.
- QST: the overall value of each sensory test is the mean at the most painful location and the five control locations together, provided that at most two locations are missing.
- To perform the analyses to respond to objective 3, delta scores are computed (difference in scores between baseline and discharge) for CS, functioning and those confounder variables which are also measured at the same time points as the main measurements.

Proxies of the main measurements were selected to perform analyses. CS variables are: the sum score of CSI part A and the RMSSD for HRV. For QST, no variables are predefined, variables that correlate most strongly with each of the functioning measures will be selected. Functioning variables are: the maximal weight lifted for lifting capacity, the $VO_2max/kg$ for the maximal aerobic capacity, the daily vector magnitude (VM) counts for PA and the average score of Rand36-PF for reported physical functioning.

The presence of missing data will be checked. If a variable has more than 10% of missing data cases, its distribution will be further examined. When the distribution of missing data is missing completely at random, analyses with pairwise deletions are planned. If data are not missing completely at random, other suitable solutions will be applied such as multiple imputation.[91] Regarding the presence of outliers, cases at 3 SD from the mean will be considered as outliers. Their influence will be measured using Cook's distance and leverage values. When the outliers are not statistically influential and/or there is rationale for the outlier, no further actions are planned. The descriptive statistics of the sample characteristics and main test results will be at this point calculated; numerical variables as mean and SD in normally distributed data and as median and IQR (25–75)) in non-normally distributed data, and categorical variables as counts and percentage of each category level.

▶ *Statistical approach for objective 1: to analyse the association between instruments assessing reference symptoms and signs attributed to CS.* Multiple partial correlation analyses will be performed and corrected for potential confounders. CS measurements will be the main variables and sex and age will be covariates. Additional covariates may enter the partial correlations based on their correlation coefficient with CS instruments and if p<0.01. Results will be reported as correlation coefficients (Pearson's r and/or Spearman's rho) and p values, with and without controlling for covariates. The significance level for all analyses is established at 0.01, to account for the potential type I error due to the multitude of partial correlation analyses.

▶ *Statistical approach for objective 2: to analyse whether reference symptoms and signs attributed to CS are associated with functioning measurement outcomes.* First, multiple correlation analyses, followed by multiple regression analyses with data obtained at baseline, will be performed. Each of the main outcomes from functioning measurements will be introduced in the models as dependent variable, CS measurements as independent variables, and sex, age and other demographic characteristics and/or test-related variables as potential confounders and/or moderators.

▶ *Statistical approach for objective 3: to analyse whether changes (between baseline and discharge) in reference symptoms and signs attributed to CS are related to changes in each of the functioning measurement outcomes.* A similar approach to objective 2 will be performed: first, multiple correlation analyses will be performed with data obtained at baseline and discharge, followed by multiple correlation analyses using delta scores of the data and, finally, multiple regression analyses again using delta scores. As in objective 2, each of the main outcomes from functioning measurements will be introduced in the models as dependent variable, CS measurements as independent variables, and sex, age and other demographic characteristics and/or test-related variables as potential confounders and/or moderators.

Confounders entering the final models of objectives 2 and 3 will be selected based on their partial correlation coefficient with CS and functioning, and they will be included if p≤0.10. All multiple regression models will have the assumptions checked. Results of objectives 2 and 3 will be reported as coefficient (b), CIs (95% CI), p values and explained variance, with and without controlling for confounders and/or moderators. Due to the large amount of correlation analyses and the risk of high type I error, a stricter significance level for correlation analyses is established, p<0.01. For multiple regression analyses the significance level is at 0.05.

## DISCUSSION

Several patients with CLBP have shown common manifestations with other chronic pain conditions, which may evidence the presence of CS in these patients. Yet, knowledge on CS, especially in patients with CLBP, and on its association with functioning is developing. This research protocol aims to bridge that gap of knowledge with a novel approach. In the first place, the study (objective 1) will describe the mechanisms of CS in patients with CLBP using instruments that assess reference symptoms of CS from different perspectives. Furthermore, the study (objectives 2 and 3) will also analyse longitudinally whether the presence of CS and changes of CS can be associated with changes in physical functioning performance, for which functioning will be assessed with several objective monitoring and capacity tests, and self-reported measurements.

Within the constraints of CAU and in the absence of a single gold standard instrument that can diagnose CS, in this research protocol multiple instruments have been selected to examine reference symptoms and signs attributed to CS. (1) The hypersensitivity unrelated to mechanical stimulus is assessed with CSI. A questionnaire designed to measure CS, as it has been shown to be associated to factors known to contribute to CS and is able to identify somatic and emotional symptoms related to CSS.[24 38 92] (2) The disproportionate pain, diffuse distribution, and allodynia and hyperalgesia are measured with QST. The QST is a battery of non-invasive standardised tests that assists in the identification of sensory and pain thresholds, the determination of pain intensity perception and the assessment of the CPM. Its results can allow pain characterisation (phenotyping) and processing, and monitoring its progression.[93–97] (3) The low vagal nerve activity is examined with HRV. A non-invasive standardised measure that characterises the function of the ANS.[34] The somatic and emotional symptoms related to CSS may generate an imbalance between a hyperactivation of the sympathetic tone and a reduction of the parasympathetic/vagal tone.[34 98 99] (4) Psychosocial factors are measured with psychological screening questionnaires for perceived injustice (IEQ), catastrophising (PCS) and psychological distress (BSI). These psychosocial factors have shown a consistent effect on functioning and on

symptom beliefs and pain behaviours, which are common in patients with chronic pain, including CSS. Therefore, although not necessary for the clinical classification of CS, these factors are included in the study and potentially in the analyses. Some of the CS assessment instruments used in this study are more acknowledged than others in CS literature. CSI is widely used to measure CS since it was designed as a proxy for it; however, its internal consistency and reliability in patients with CLBP has not yet been made available. Also, evidence is currently piling up on the necessity of routinely implementing the elements of QST relevant to patients' conditions in clinical practice, given its ability to assess the somatosensory function,[97 100] even though evidence in patients with CLBP is limited. Finally, literature on HRV as an instrument that may reflect an imbalance of ANS in patients with chronic pain conditions (specifically in those patients who have shown an overlap in the CS symptoms) is scarce.[101] We are confident that the selected CS assessment instruments are considered the best available and combined will be able to comprehensively measure from the different reference clinical symptoms and signs indicative of CS being present. This will provide further insights on the mechanisms of CS and assist on the phenotyping of centrally sensitised patients with CLBP.

Various functioning assessments are selected for the research protocol to determine the functioning status of centrally sensitised patients with CLBP, including several objective monitoring and capacity tests, and self-reported measurements. (1) The floor-to-waist lifting capacity test is a physical assessment from the FCE that has demonstrated the best predictability for return to work in CLBP.[102] (2) The aerobic capacity, expressed as VO$_2$max, can be best assessed with a maximal aerobic capacity test such as the CPET. The cycle ergometer and the treadmill are the most used assessments for aerobic capacity, but the cycle ergometer is most appropriate for patients.[37] (3) The PA is measured by an accelerometer, an easy wearable activity-monitoring device that provides objective measurements of PA levels and PA distribution or patterns. (4) Patient-reported functioning questionnaires show patients' pain experience, how they perceive or believe their functioning and participation is. The WAS, PDI and Rand36-PF show the advantage of being of short length and easy to interpret, due to their interval scale. Although questionnaire results can differ from their actual physical performance, both measurements complement each other. Furthermore, together they may provide a more comprehensive overview of patients with CLBP.[103–105] Assessments such as lifting capacity and self-reported measurements have been routinely used to determine the functioning status of patients with CLBP. CPET, on the other hand, despite being scientifically superior to a submaximal aerobic capacity test, has barely been implemented with patients with CLBP. As a result, it is unknown whether a maximal CPET could be applied systematically with these patients and their pain response to it. Moreover, even if there is a large amount of literature on PA, the methodology implemented is very broad regarding population

groups and/or measuring instruments, which hinders the synthesis of the understanding of PA of patients with CLBP. To bridge the gap, it may be of use an objective measurement of PA and monitored with accelerometers such as those of ActiGraph, which has shown to produce the most reliable devices for recording PA in free-living conditions.[106 107] It is believed that the assessments described are suitable and relevant for the purpose of measuring functioning from multiple domains; and that the extended knowledge and the better understanding of the mechanisms involved obtained is not limited to CLBP but can be extended to CS.

Data collected for this project are large and the measurements of CS and functioning collected involve a broad amount of variables. As a result, proxies will be used for the statistical analyses. Previous research in CS assessments, on the one hand, and in functioning assessments, on the other, have reported different methodologies and outcome parameters. The reason for which, even if authors have selected the proxy variables based on their relevance and/or contribution to the purposes of the current studies, the selection of the following measurements as proxies may not be beyond debate: for HRV the RMSSD, as it is a measurement of the parasympathetic/ vagal tone which is not influenced by breathing[108]; for the maximal aerobic capacity the VO$_2$max corrected for the body mass of the individual, due to the effect of body mass on VO$_2$max; and for PA the daily VM counts, because it characterises movement duration and intensity.[109]

There are circumstances that might have influenced the results, in spite of the team's efforts to obtain the most extensive insights of CS and its association with functioning. To begin with, it should be considered that patients participating in the study are volunteers. Moreover, the exclusion criteria, despite being based on patients' files, are also dependent on doctor's judgement. The impact of these circumstances is so far unknown to the authors. Likewise, given the CAU design of the study the choice of measurements has to be feasible and compatible with the regular clinical procedures. As a consequence, the functional MRI, a commonly mentioned instrument to measure CS, could not be performed in the study. The HRV, on the other hand, although less often used, can be an insightful and innovative approach to the assessment of one of the indicators of CS. Additionally, the sample sizes are estimated a priori for the analyses due to insufficient previous studies and, therefore, they are of exploratory nature and used as a guide. Also, despite the efforts of the team to promote the study participation and participant retention, it is likely that patient recruitment may be slower than anticipated and that the patient may not have a complete follow-up. However, since the measurement period is ample, the calculations are vigilantly made with generous margins (see the Methods and analyses section, Sample size), and the literature related to the topic has shown smaller sample sizes.[25 28 110] We remain positive about the sample size and expect to have more than enough participants to perform analyses with sufficient power. Also, for comparability purposes, measurements

are planned to take place within 3 weeks from the Pain Rehabilitation team's baseline and discharge assessments. The CSI, PDI and psychological screening questionnaires during baseline assessment are sent to be filled up 6 weeks before the rest of the measurements are obtained, a time distance due to waiting list time. Measurements are expected to be comparable to the rest of measurements at baseline, since no important changes are foreseen in patients regarding medication, surgery or personal conditions during this time, and questionnaires have shown to have good clinimetric properties. As opposed to the limitations presented, given the large amount of correlations in the multiple correlation analyses and the risk of type I error, a more strict significance level was applied, that is, 0.01. Also, for the QST the proxy will be selected per functioning outcome to obtain the most informative somatosensory measurement for each of the functioning measurements. We are confident that the results obtained will provide meaningful and valuable information.

All things considered, the results obtained from the measurements described in this research protocol can, first, help better describe and understand the mechanisms involved in CS, particularly the ones affecting the functioning of patients with CLBP; and second, can be used to optimise the effectiveness of pain management interventions. To the authors' knowledge, the study described in this research protocol is the first of its kind and it may become an important breakthrough for researchers involved in the study of pain, for the treatment strategy of patients with chronic pain by healthcare professionals and for patient's pain education.

## ETHICS AND DISSEMINATION

All participating patients sign the informed consent before any assessment takes place. Formal ethical clearance to perform the study has been obtained from the Medical Research Ethics Committee (METc) of the University Medical Center Groningen (the Netherlands) in July 2017 (METc 2016/702). All procedures are in accordance with the ethical standards of the Helsinki Declaration of 1975 as revised in 2014[111] and with Dutch regulations of the Medical Research Involving Human Subjects Act, in Dutch: Wet Medisch-Wetenschappelijk Onderzoek, amended in 2018.[112]

The dissemination strategy of the research study results will be through scientific publications in peer-reviewed journals and presentations in relevant national and international conferences. In addition to it, reports will be sent by post to participants, and reports and presentations in meetings will be provided to stakeholders.

**Contributors** JAE was responsible for the study concept and design, data collection, and manuscript preparation and edit. HRSP, RD and MFR were responsible for the study concept and design, and critical review of the manuscript. IS, HT and APW were responsible for the study design and critical review of the manuscript.

**Funding** The Pain Rehabilitation Development Center (in Dutch: Ontwikkelcentrum PijnRevalidatie (OPR)) of the CvR-UMCG funded the research study.

**Competing interests** None declared.

**Patient consent for publication** Not required.

**Ethics approval** The Medical Research Ethics Committee of the University Medical Center Groningen (METc UMCG) provided a positive reaction in July 2017 to the development of the research study 'Central Sensitization and Functioning in patients with Chronic Low Back Pain' (METc 2016/702). Its first amendment was approved in October 2017 and its second amendment in August 2018.

**Provenance and peer review** Not commissioned; externally peer reviewed.

**ORCID iD**
Jone Ansuategui Echeita http://orcid.org/0000-0002-7577-4348

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
