## [Reviewer comments · BMJ Open]

ARTICLE DETAILS

TITLE (PROVISIONAL)	Central Sensitisation and Functioning in patients with Chronic Low Back Pain: Protocol for a cross-sectional and cohort study
AUTHORS	Ansuategui Echeita, Jone; Schiphorst Preuper, Henrica R; Dekker, Rienk; Stuive, Ilse; Timmerman, Hans; Wolff, Andre P; Reneman, Michiel Felix

VERSION 1 – REVIEW

REVIEWER	Samantha Meints Brigham and Women's Hospital, USA
REVIEW RETURNED	10-Jun-2019

GENERAL COMMENTS	Abstract 1. It is unclear if CS is your predictor or your outcome for the analyses in sections 2 and 3. Introduction 2. On page 5 the authors provide two possible reasons that an association between CS and physical function may exist. However, it is unclear what the authors are trying to say, in part because this is wordy and the grammar is poor. This section is overall quite cumbersome to decipher. 3. The authors indicate that there are no known associations between CS and physical function. However this is not true. Indeed, there is a 2013 systematic review and meta-analysis on the topic (Hubscher et al., 2013). 4. On page 5, lines 16-20, authors indicate one purpose of this study is to develop a diagnostic instrument for CS. However, that seems outside the scope of this project. Objectives 5. Throughout the introduction, there lacks a clear rationale for the objectives identified here. Specifically, why these measures? What is special about CSI, QST, HRV? Why are these or should these be associated with lifting capacity, aerobic capacity, PA levels, etc? Participants 6. Are participants with other pain comorbidities excluded? What about opioid use? If not, is this controlled for during any analyses? Procedures 7. There is no rationale for inclusion of the psychological screening questionnaires. This is necessary. For example, why catastrophizing and injustice and not fear of movement and pain acceptance? Sample Size 8. You need some previous effect size or correlation coefficient for your expected correlation between these measures of CS. If these all are presumed to measure CS, I would expect larger effect sizes
---

	than just .3. Moreover, will you control for any other variables such as pain intensity which can influence CS? 9. The “rule of thumb” of 10 participants per variable is unacceptable. This needs more detail. Data Analysis 10. Why use listwise deletion for missing variables rather than pairwise? What will you do if the data is no MCAR? 11. How will you handle non-normal data? 12. Overall, your decisions here should be backed by references from statistics literature. 13. How will you decide what to keep in the models? How will you account for type I error given the multitude of analyses? How will you address any interactions?
--	--

REVIEWER	Assoc Prof Donna Urquhart Monash University, Australia
REVIEW RETURNED	16-Jul-2019

GENERAL COMMENTS	Manuscript ID: BMJopen-2019-031592 Manuscript Title: Central Sensitization and Functioning in patients with Chronic Low Back Pain: Protocol for a cross-sectional and cohort study Background: This protocol indicates that a limited number of studies have examined the relationship between instruments that assess central sensitisation, and the association between central sensitisation and physical function. Despite this, no information is provided regarding previous work and how the proposed study targets gaps in our scientific understanding. A brief search of the literature identified the following studies which have investigated the proposed study’s aims. Thus, I would suggest a more comprehensive review of the current studies in the area is required and that this be included in the introduction and the discussion. References:  1. Smart KM; Blake C; Staines A; Thacker M; Doody C. Mechanisms-based classifications of musculoskeletal pain: part 1 of 3: symptoms and signs of central sensitisation in patients with low back (+/- leg) pain. Manual Therapy. 17(4):336-44, 2012 Aug. 2. Smart KM; Blake C; Staines A; Doody C. Self-reported pain severity, quality of life, disability, anxiety and depression in patients classified with 'nociceptive', 'peripheral neuropathic' and 'central sensitisation' pain. The discriminant validity of mechanisms-based classifications of low back (+/-leg) pain. Manual Therapy. 17(2):119-25, 2012 Apr. Aims: The protocol presents 3 aims. However, it is not clear how Aim 2 and 3 are different. Both are looking at the association between central sensitisation and physical functioning. The first is examining whether there is an association and the second is looking at the direction of the association – which will be answered by meeting the first aim. I would suggest combining these aims in to one. The protocol uses the word “trial” in the manuscript. I think this is confusing as this study is not a clinical trial rather an observational study. I would suggest using the word “study” instead. Methods: The protocol outlines the study inclusion and exclusion criteria. However, I have concerns that these are quite broad. For instance, there is no mention of excluding participants that have low back pain due to pathological reasons, such as cancer,
--

	osteoporosis, and spinal fractures. Do the authors wish to focus this study on all types of low back pain or rather non-specific low back pain with the exclusion of pathological low back pain? It is important to consider this question as low back pain is a heterogeneous condition and it is well accepted that low back pain due to pathological causes is very different to non-specific low back pain. Moreover, what about patients that have had surgery or plan to have low back pain surgery? The inclusion and exclusion criteria need to be reconsidered. There is a significant amount of information provided on each of the instruments to be used, including their specificity and sensitivity. However, limited information is presented that justifies the choice of the instruments relative to other instruments that are available. I would suggest further information is provided to justify the choice of instruments. It is planned that some of the data will be collected 6 weeks prior to baseline and other data will be collected at baseline. Together, these data will be considered “baseline” data. However, there are issues associated with this as participants’ low back pain may significantly change during the 6 weeks period prior to baseline. Why not collect all the baseline data at one time period? Statistical analyses: There are a few aspects of the statistical analyses that require attention. Firstly, it is proposed that multiple correlations are to be performed. However, there is no mention of the possible issues with this and how they might be overcome. Secondly, given there is limited studies in this field, the authors propose that their sample size is based on the number of variables to be included in the analyses. However, this method has several limitations and in addition, a search identifies available studies. Strength/limitations section (Pg 3). It is unclear what the following means: “Statistical analyses are performed with proxies of CS and functioning measurements based on their relevance and contribution; the decision may not be beyond debate.
--	---

VERSION 1 – AUTHOR RESPONSE

Author’s response to Reviewer #1, dr. Samantha Meints, comments:

Abstract

1. It is unclear if CS is your predictor or your outcome for the analyses in sections 2 and 3.

Response: Thank you for the comment. We have added to the analysis part which variables are the dependent and which the independent.

***Changes to the manuscript:* page 2 lines 15-17.**

Introduction

2. On page 5 the authors provide two possible reasons that an association between CS and physical function may exist. However, it is unclear what the authors are trying to say, in part because this is wordy and the grammar is poor. This section is overall quite cumbersome to decipher.

Response: Theoretically we expect that CS and functioning are associated in individuals with CLBP, because CS can be present in individuals with CLBP and functioning is limited in individuals with CLBP. Due to the increased responsiveness to stimuli when CS is present, individuals with CLBP which are affected by CS are likely to have their functioning more limited. This is the

hypothesis we are aiming to test with our study protocol. We have further developed the line of reasoning in the paragraph of the Introduction section.

Changes to the manuscript: page 5 lines 13-22.

3. The authors indicate that there are no known associations between CS and physical function. However this is not true. Indeed, there is a 2013 systematic review and meta-analysis on the topic (Hubscher et al., 2013).

Response: Thank you very much for mentioning this interesting review paper. The review aimed to examine the relationship between QST measures and reported pain or disability, the aim of our research protocol differs from the review of Hubscher et al. because we aim to study CS and functioning.

In our study we collect a wider amount of CS measures compared to the review, where only QST measures were searched. Another difference is that we study functioning, which is related to but not the same as disability. ICF defines functioning as: “body functions, body structures, activities and participation”; and disability as “impairments, activity limitations and participation restrictions” [WHO 2013]. The review focused on disability and, although it includes 2 papers on reported physical functioning these were not accounted differently from disability in their association with QST. Additionally, in our study we have questionnaires and performance-based measures for functioning, which did not appear in the review.

To our knowledge evidence that assesses the association between CS and functioning as comprehensively as we aim to is not known. The associations that have been described in literature are very restricted and limited to certain instruments. We have emphasized the scarcity in CS and functioning literature in the Introduction section and supported the statement with the paper of Hubscher et al..

Reference:

World Health Organization. How to use the ICF: A practical manual for using the International Classification of Functioning, Disability and Health (ICF). Exposure draft for comment. October 2013. Geneva: WHO

Changes to the manuscript: page 5 lines 4-10 and 22-24.

4. On page 5, lines 16-20, authors indicate one purpose of this study is to develop a diagnostic instrument for CS. However, that seems outside the scope of this project.

Response: We agree with the reviewer. For clarification we have deleted the part of the sentence referring to the lack of diagnostic instrument for CS and further explained the gaps in knowledge regarding CS and functioning literature.

Changes to the manuscript: page 5 line 23.

Objectives

5. Throughout the introduction, there lacks a clear rationale for the objectives identified here. Specifically, why these measures? What is special about CSI, QST, HRV? Why are these or should these be associated with lifting capacity, aerobic capacity, PA levels, etc?

Response: CSI, QST, and HRV assess indicators of CS. Due to the lack of gold standard to diagnose CS, only symptoms and signs related to it may be identified. These 3 instruments are chosen because each of them assesses different aspects of CS which can also be complementary to describe CS, and also because of their feasibility and clinimetric properties.

Within the different domains in the functioning and participation components, lifting capacity, Physical Activity (PA) levels and/or PA distribution or patterns, aerobic capacity, and daily activities participation; are representative of the mobility, self-care, work and leisure performance, and essential for patients with LBP [Cieza et al.2004].

We have further developed the paragraphs CS and functioning in relation to their measures, in order to clarify the measurements and to be better linked to the objectives presented.

Reference:

Cieza A, Stucki G, Weigl M, Disler P, et al. ICF Core Sets for low back pain. *J Rehabil Med* 2004; suppl. 44: 69–74.

Changes to the manuscript: page 4 lines 22-25 and page 5 lines 1-3.

Participants

6. Are participants with other pain comorbidities excluded? What about opioid use? If not, is this controlled for during any analyses?

Response: *Participants with other pain comorbidities are not excluded because various comorbid symptoms and conditions co-occur frequently in individuals with CS. The exclusion of potential participants due to pain comorbidities would notably endanger the inclusion of the targeted population for the study.*

The use of opioid medication is neither a criterion for exclusion and such information is recorded. Although it has not been mentioned specifically in the analysis section, the use of medication and the presence or absence of comorbidities, will enter the multiple correlation analyses and may be selected as confounder for the multiple regression analyses; depending on their partial correlation coefficient with functioning outcomes and provided it fulfills with the $p \leq 0.10$ requirement (see Methods Section, page 17 line 19-20).

Changes to the manuscript: *We have further specified that pain comorbidities are not exclusion criteria of participants for the study; page 7 lines 10-11.*

Procedures

7. There is no rationale for inclusion of the psychological screening questionnaires. This is necessary. For example, why catastrophizing and injustice and not fear of movement and pain acceptance?

Response: *There are several cognitive and emotional features common to patients with chronic pain that affect their clinical presentation. PCS and IEQ (catastrophizing and perceived injustice, respectively) measure features that have been associated to CS syndromes (Adams and Turk 2015). Catastrophic thinking affects pain perception and experience, and could be incorporated to the fear avoidance model, e.g. higher catastrophic thoughts may be related to more fear and more movement avoidance. Perceived injustice may be expressed as anger and frustration; where more anger expression has been shown to be related to lower endogenous inhibitory mechanisms and increased muscle tension leading to more pain experience.*

Due to the relation of catastrophizing with the experience of pain and fear of movement, we believe that PCS provides the necessary insights on this feature. And because perceived injustice is related to the accentuation of CS symptoms through anger and frustration feelings, we think IEQ provides more interesting and specific features of the coping strategy of patient to the phenomena we aim to study.

We have provided a short rationale in the methods section.

Reference:

Adams LM, Turk DC. Psychosocial Factors and Central Sensitivity Syndromes. *Curr Rheumatol Rev.* 2015; 11(2): 96–108.

Changes to the manuscript: page 14 lines 10-11.

Sample Size

8. You need some previous effect size or correlation coefficient for your expected correlation between these measures of CS. If these all are presumed to measure CS, I would expect larger effect sizes than just .3. Moreover, will you control for any other variables such as pain intensity which can influence CS?

Response: We understand that a correlation coefficient of 0.3 might seem somewhat low. CSI, QST, and HRV assess different aspects of CS and from very different perspectives: patient-reported, somatosensory function, and Autonomous Nervous System function. Due to this, it is deemed unlikely that these instruments will strongly relate to each other and we do not expect the individual correlations to be higher than 0.3. Because there are no previous studies reporting correlations between these instruments; we used this estimate of 0.3 for our sample size calculation.

Regarding, pain intensity, it will enter the multiple correlation analyses and may be selected as confounder for the multiple regression analyses; provided it fulfills with the requirements described in the Methods section (see page 16 lines 26-27 and page 17 lines 16-17).

Changes to the manuscript: page 16 lines 28-29.

9. The “rule of thumb” of 10 participants per variable is unacceptable. This needs more detail.

Response: We agree with the reviewer that the proposed ‘Rule of thumb’ is not ideal for sample size estimation, but we currently don’t know of any better alternative. There are several ‘Rule of thumb’-s available for multiple linear regression analyses, from which a minimum of 10 events per independent variable entered in the model seems to be sufficient requirement [Wilson van Voorhis and Morgan 2007]. Because the study is exploratory, there is no previous evidence on the association between CS and functioning which we could use to estimate a sample size.

Reference:

Wilson van Voorhis CR, Morgan BL. Understanding power and rules of thumb for determining sample sizes. *Tutorials in quantitative methods for psychology*. 2007; 3(2): 43-50.

Changes to the manuscript: none.

Data Analysis

10. Why use listwise deletion for missing variables rather than pairwise?

Response: We agree with the reviewer in the use of pairwise deletion in the correlation analyses it is more appropriate. We have adapted it in the manuscript accordingly.

Changes to the manuscript: page 16 line 16.

11. What will you do if the data is no MCAR?

Response: Please see response to Item 13.

12. How will you handle non-normal data?

Response: Please see response to Item 13.

13. Overall, your decisions here should be backed by references from statistics literature.

Response: The decision of how should be handled the MAR and NMAR values and non-normal data will be decided depending on the specific characteristics of the database. We aim to keep any adaptation to a minimum while maintaining the statistical power according to the specific characteristics of the data available and to fully report the approach applied.

MAR and NMAR values are likely to undergo imputation [van Ginkel et al. 2018]; and non-normally distributed data will be analyzed with a non-parametric method when available, such as Spearman's rank correlation for the multiple correlations. If no non-parametric method meeting the study's objectives is available, may follow transformation by categorization and/or log-transformation.

We have added the description of non-normally distributed data (median and interquartile range) to the presentation of the characteristics of the sample, we also have made a suggestion on how to handle no MCAR missing data, and added the type of correlation coefficients that could be expected from the analyses.

Reference:

van Ginkel JR, Linting M, Rippe RC, van der Voort A. Rebutting Existing Misconceptions About Multiple Imputation as a Method for Handling Missing Data. *Journal of personality assessment*. 2019; 1-12.

Changes to the manuscript: page 16 lines 15-17, 21-22 and 28.

14. How will you decide what to keep in the models? How will you account for type I error given the multitude of analyses? How will you address any interactions?

Response: We agree with the reviewer that there are a multitude of correlation analyses to be done. The idea of performing Bonferroni adjustments has been considered for the multiple correlation analyses, but:

- 1) Due to the large amount of correlations in the multiple correlation analyses, a Bonferroni adjustment would lead to an increase in Type II error when trying to avoid Type I error.
- 2) Bonferroni would only adjust for the significance. Establishing results and conclusions only on significance can be flawed [Ziliak and McCloskey 2008]; moreover, we are interested in the size of effect of the associations.

All things considered we have decided to have a stricter significance level for multiple correlation analyses, i.e. 0.01; and to provide 95% Confidence Intervals of the correlation coefficients in the results as they can be more insightful of the values of the population. We believe that with these measures we will be able to extract meaningful information.

For the interactions, the variables that have shown to be confounders in the multiple regression models will be explored for possible interaction/moderation effect, provided the variables are theoretically and statistically sound.

Reference:

Ziliak S, McCloskey DN. *The cult of statistical significance: How the standard error costs us jobs, justice, and lives*. University of Michigan Press; 2008.

Changes to the manuscript: page 16 line 29 and page 17 line 20.

Author's response to Reviewer #2, Assoc. Prof. Donna Urquhart, comments:

Background

15. This protocol indicates that a limited number of studies have examined the relationship between instruments that assess central sensitisation, and the association between central sensitisation and physical function. Despite this, no information is provided regarding previous work and how the proposed study targets gaps in our scientific understanding. A brief search of the literature identified the following studies which have investigated the proposed study's aims. Thus, I would suggest a more comprehensive review of the current studies in the area is required and that this be included in the introduction and the discussion.

References:

1. Smart KM; Blake C; Staines A; Thacker M; Doody C. Mechanisms-based classifications of musculoskeletal pain: part 1 of 3: symptoms and signs of central sensitisation in patients with low back (+/- leg) pain. *Manual Therapy*. 17(4):336-44, 2012 Aug.
2. Smart KM; Blake C; Staines A; Doody C. Self-reported pain severity, quality of life, disability, anxiety and depression in patients classified with 'nociceptive', 'peripheral neuropathic' and 'central sensitisation' pain. The discriminant validity of mechanisms-based classifications of low back (+/-leg) pain. *Manual Therapy*. 17(2):119-25, 2012 Apr.

Response: Thank you for your comment. Research in CS is scarce, and the associations described in literature between CS instruments and between CS and functioning are very restricted and limited to certain instruments. Studies on CS like the ones mentioned have been key to describe a criteria for the identification of different clinical manifestations of CS (see Introduction section page 4 paragraph 2), based on which specific instruments have been developed to assess the underlying mechanisms of CS. We aim to use some of the instruments designed to measure the symptoms and signs of CS in order to assist in the profiling of CS, from a broad perspective and specifically for patients with CLBP, and to relate the outcomes to various aspects of functioning. To our knowledge evidence that assesses the association between CS and functioning as comprehensively as we aim is not known.

We have emphasized the scarcity in CS and functioning literature in the Introduction section, and supported the symptoms and signs of CS in the Introduction section with the paper of Smart et al. 2012.

Changes to the manuscript: page 5 lines 9-10 and lines 22-25.

Aims

16. The protocol presents 3 aims. However, it is not clear how Aim 2 and 3 are different. Both are looking at the association between central sensitisation and physical functioning. The first is examining whether there is an association and the second is looking at the direction of the association – which will be answered by meeting the first aim. I would suggest combining these aims in to one.

Response: The main difference between objective 2 and 3 are the study designs.

- *Objective 2: a cross-sectional study. Analyses the association between CS and functioning using the baseline scores.*
- *Objective 3: a longitudinal study. Analyses the influence of CS on functioning, where the change in scores (calculated by subtracting baseline from discharge scores) for both CS and functioning variables are used.*

We have tried to be clearer in the study objectives, and changed the manuscript accordingly.

Changes to the manuscript: page 2 lines 7-10, page 6 lines 5-16, page 15 lines 13-17, page 17 lines 1-2, 7-9 and 28-29.

17. The protocol uses the word “trial” in the manuscript. I think this is confusing as this study is not a clinical trial rather an observational study. I would suggest using the word “study” instead.

Response: Thank you for the suggestion, we have changed the text accordingly.

Changes to the manuscript: page 6 line 19.

Methods

18. The protocol outlines the study inclusion and exclusion criteria. However, I have concerns that these are quite broad. For instance, there is no mention of excluding participants that have low back pain due to pathological reasons, such as cancer, osteoporosis, and spinal fractures. Do the authors wish to focus this study on all types of low back pain or rather non-specific low back pain with the exclusion of pathological low back pain? It is important to consider this question as low back pain is a heterogeneous condition and it is well accepted that low back pain due to

pathological causes very different to non-specific low back pain. Moreover, what about patients that have had surgery or plan to have low back pain surgery? The inclusion and exclusion criteria need to be reconsidered.

Response: We agree that LBP is a very heterogeneous condition. For our study we aim to include patients whose primary diagnosis is LBP. According to the ICD-11, chronic primary musculoskeletal pain is a chronic pain whose symptoms are not better accounted for by any other diagnosis and are associated to important functional interference and emotional distress [IASP Taskforce for the Classification of Chronic Pain 2019].

Therefore, patients participating in the study must not be diagnosed with LBP due to other diagnoses as cancer, osteoporosis, and/or spinal fractures; and/or be appointed for a LBP surgery, because in those cases a different cause for the CLBP has been identified. Nevertheless, participants could have undergone a surgery in the past, but such must not be the cause for their current chronic pain. It is our aim that no specific pathological process has been identified or is supposed to be present clarifying the origin of the pain of the participating patients.

Following your remarks, we have made the paragraph more explanatory and provided examples for the exclusion criteria.

Reference:

IASP Taskforce for the Classification of Chronic Pain. The IASP classification of chronic pain for ICD-11: chronic primary pain. Pain. 2019. 160(1); 28-37.

Changes to the manuscript: page 7 line 1-11.

19. There is a significant amount of information provided on each of the instruments to be used, including their specificity and sensitivity. However, limited information is presented that justifies the choice of the instruments relative to other instruments that are available. I would suggest further information is provided to justify the choice of instruments.

Response: CSI, QST, and HRV assess indicators of CS. Due to the lack of gold standard to diagnose CS, only symptoms and signs related to it may be identified. These 3 instruments are chosen because each of them assesses different aspects of CS which can also be complementary to describe CS, their feasibility, and their clinimetric properties.

Within the different domains in the functioning and participation components, lifting capacity, Physical Activity (PA) levels and/or PA distribution or patterns, aerobic capacity, and daily activities participation; are representative of the mobility, self-care, work and leisure performance and essential for patients with LBP [Cieza et al.2004].

We have further developed the paragraphs CS and functioning in relation to the measures, in order to clarify the measurements and to be better linked to the objectives presented.

Reference:

Cieza A, Stucki G, Weigl M, Disler P, et al. ICF Core Sets for low back pain. J Rehabil Med 2004; suppl. 44: 69–74.

Changes to the manuscript: page 4 lines 22-25 and page 5 lines 1-3.

20. It is planned that some of the data will be collected 6 weeks prior to baseline and other data will be collected at baseline. Together, these data will be considered “baseline” data. However, there are issues associated with this as participants' low back pain may significantly change during the 6 weeks period prior to baseline. Why not collect all the baseline data at one time period?

Response: Thank you for the comment. The whole study is performed as part of Care As Usual for outpatients, which means that we are connected to regular clinical procedures. However, we do not expect relevant differences in the outcomes due to the 6 week's waiting time, because: all those questionnaires have shown good or excellent test-retest reliability and patients should follow

on their daily routines with no changes during this period of time regarding their behavior, including medication, surgery, or personal conditions.

We have more extensively explained this limitation in the Discussion section.

Test-retest reliability:

CSI: ICC = 0.91-0.88; PDI: ICC = 0.76; PCS: $r = 0.92$; IEQ: ICC = 0.98; BSI: $r = 0.71-0.89$.

Changes to the manuscript: page 20 lines 4-9.

Statistical analyses

21. There are a few aspects of the statistical analyses that require attention. Firstly, it is proposed that multiple correlations are to be performed. However, there is no mention of the possible issues with this and how they might be overcome.

Response: There are a large amount of multiple correlation analyses to be performed in order to respond the main objectives. We are aware that there is a great risk of type I error and the idea of performing Bonferroni adjustments has been considered, but:

- 1) Due to the large amount of correlations in the multiple correlation analyses, a Bonferroni adjustment would lead to an increase in Type II error when trying to avoid Type I error.
- 2) Bonferroni would only adjust for the significance. Establishing results and conclusions only on significance can be flawed [Ziliak and McCloskey 2008]; moreover, we are interested in the size of effect of the associations which would not change even if the significance is adjusted. In this situation and all things considered to have a stricter significance level for multiple correlation analyses, i.e. 0.01; and to provide 95% Confidence Intervals of the correlation coefficients in the results because they can be more insightful of the values of the population. We believe that with these measures we will be able to extract meaningful information.

Reference:

Ziliak S, McCloskey DN. *The cult of statistical significance: How the standard error costs us jobs, justice, and lives.* University of Michigan Press; 2008.

Changes to the manuscript: page 16 line 29 and page 17 line 20.

22. Secondly, given there is limited studies in this field, the authors propose that their sample size is based on the number of variables to be included in the analyses. However, this method has several limitations and in addition, a search identifies available studies.

Response: We agree with the reviewer that the 'rule of thumb' is not ideal for sample size estimation; unfortunately, we don't know of any better alternative. There are several 'Rule of thumb'-s available for multiple linear regression analyses, from which a minimum of 10 events per independent variable entered in the model seems to be sufficient requirement [Wilson van Voorhis and Morgan 2007]. The study is exploratory and even if wanted there is no previous evidence on the association between CS and functioning which we could use to estimate a sample size. The studies proposed by the reviewer might seem similar but their aim is different from the current study (see response to Item 15).

Reference:

Wilson van Voorhis CR, Morgan BL. *Understanding power and rules of thumb for determining sample sizes.* *Tutorials in quantitative methods for psychology.* 2007; 3(2): 43-50.

Changes to the manuscript: none.

23. Strength/limitations section (Pg 3). It is unclear what the following means: "Statistical analyses are performed with proxies of CS and functioning measurements based on their relevance and contribution; the decision may not be beyond debate".

Response: We selected measures as operationalizations for CS and functioning from previously published papers on CS and functioning. The selection of the measures is made to the best of our knowledge of the literature to meet the study's aims, but we cannot discard the possibility that other selections would lead to different results.

Following the reviewer's remark, we have made the statement more explanatory.

Changes to the manuscript: page 3 line 8-10.

VERSION 2 – REVIEW

REVIEWER	Samantha Meints Brigham and Women's Hospital, Harvard Medical School, USA
REVIEW RETURNED	05-Sep-2019

GENERAL COMMENTS	I commend the authors in their addressing reviewer concerns. However, consistent with reviewer #2, the rule of thumb is not sufficient for the sample size estimate. As noted by reviewer #2 as well, you should identify similar studies to determine estimated effect sizes and use this to complete your sample size calculation. As you noted, there are differences between what has already been published in this area and what you are proposing. However, these differences can be noted in the manuscript to suggest that the sample sizes used are just a guide.
---

REVIEWER	Assoc Prof Donna Urquhart Monash University, Australia
REVIEW RETURNED	30-Sep-2019

GENERAL COMMENTS	Thanks to the authors for considering the feedback and make changes to manuscript based on the suggestions provided. While these have improved the quality and content of the manuscript, there are issues that need further comment. I would suggest that the authors include their changes to the manuscript within the response document in future to assist the review process. 1. Background. The authors state that they performed the following: We have emphasized the scarcity in CS and functioning literature in the Introduction section, and supported the symptoms and signs of CS in the Introduction section with the paper of Smart et al. 2012. Changes to the manuscript: page 5 lines 9-10 and lines 22-25. The authors have essentially stated that the literature is scarce. However, both reviewers have referred to information in the literature on this topic and suggested that a summary of the current literature be included. I would suggest that this still needs to be addressed. Moreover, while changes were made in paragraph 4 of the introduction, there are no references provided to support the new information added. 2. Justification of choice of instruments. It is still not clear why these particular instruments have been chosen compared to other options. There needs to be a clear discussion about what other
--

	instruments are available and why instrument A was chosen over instrument B. 3. Inclusion and exclusion criteria: This is no mention of whether participants that have had or are planning surgery are included. 4. Statistical issues: The statistical issues raised in the manuscript need to be also mentioned in the limitations section of the discussion.
--	--

VERSION 2 – AUTHOR RESPONSE

Author’s response to Reviewer #1, dr. Samantha Meints, comments:

I commend the authors in their addressing reviewer concerns. However, consistent with reviewer #2, the rule of thumb is not sufficient for the sample size estimate. As noted by reviewer #2 as well, you should identify similar studies to determine estimated effect sizes and use this to complete your sample size calculation. As you noted, there are differences between what has already been published in this area and what you are proposing. However, these differences can be noted in the manuscript to suggest that the sample sizes used are just a guide.

Response: Thank you for the proposal. We have looked at previous literature published in this topic with patients with CLBP and with a similar design. Although, there are no studies performing a multiple regression model to base on to calculate samples sizes for Objectives 2 and 3. We have based on the publication of Huysmans et al.2018 to define medium effects for the multiple regression analyses and adapted the methods and discussion accordingly.

Reference:

Huysmans E, Ickmans K, Van Dyck D, Nijs J, Gidron Y, Roussel N, Polli A, Moens M, Goudman L, De Kooning M. Association Between Symptoms of Central Sensitization and Cognitive Behavioral Factors in People With Chronic Nonspecific Low Back Pain: A Cross-sectional Study. *J Manipulative Physiol Ther.* 2018;41(2):92-101. doi: 10.1016/j.jmpt.2017.08.007.

Changes to the manuscript: page 14 lines 26-28, page 15 lines 12-20, and page 19 line 30 to page 20 line 6.

Author’s response to Reviewer #2, Assoc. Prof. Donna Urquhart, comments:

Thanks to the authors for considering the feedback and make changes to manuscript based on the suggestions provided. While these have improved the quality and content of the manuscript, there are issues that need further comment.

I would suggest that the authors include their changes to the manuscript within the response document in future to assist the review process.

1. Background. The authors state that they performed the following:

“We have emphasized the scarcity in CS and functioning literature in the Introduction section, and supported the symptoms and signs of CS in the Introduction section with the paper of Smart et al. 2012. Changes to the manuscript: page 5 lines 9-10 and lines 22-25.”

The authors have essentially stated that the literature is scarce. However, both reviewers have referred to information in the literature on this topic and suggested that a summary of the current literature be included. I would suggest that this still needs to be addressed. Moreover, while changes were made in paragraph 4 of the introduction, there are no references provided to support the new information added.

Response: Thank you for the comment, the issue has been fixed.

Changes to the manuscript: page 5 line 7.

2. Justification of choice of instruments. It is still not clear why these particular instruments have been chosen compared to other options. There needs to be a clear discussion about what other instruments are available and why instrument A was chosen over instrument B.

Response:

For CS, there is no gold standard diagnostic tool, different measurement instruments of CS such as questionnaires, QST and CPM, fMRI are the most commonly reported [den Boer et al. 2018]. With the exception of fMRI the rest of the measurements are included in the study. The exclusion of fMRI was due to feasibility and compatibility with the usual clinical care of the patients.

For functioning too the feasibility and compatibility with regular practices was necessary. The physical assessments are performed regularly and are the most advocated instruments to assess the functioning domains aimed. The self-reported questionnaires were chosen due to being short and easy to interpret besides being valid and reliable, making them the best choice possible.

Reference:

den Boer C, Dries L, Terluin B, et al. Central sensitization in chronic pain and medically unexplained symptom research: A systematic review of definitions, operationalizations and measurement instruments. *J Psychosom Res.* 2019;117:32–40.

Changes to the manuscript: page 17 lines 25-27, page 18 lines 24-31, and page 19 lines 27-30.

3. Inclusion and exclusion criteria: This is no mention of whether participants that have had or are planning surgery are included.

Response: Patients included follow an interdisciplinary pain rehabilitation program and are not expected to go under surgery as part of care as usual. On the other hand, if patients have gone under surgery in the past, they could still be included in the study as long as they are diagnosed of chronic primary low back pain. We have specified this last characteristic in the manuscript.

Changes to the manuscript: page 6 lines 21-22.

4. Statistical issues: The statistical issues raised in the manuscript need to be also mentioned in the limitations section of the discussion.

Changes to the manuscript: page 19 line 30 to page 20 line 6 and page 20 lines

VERSION 3 – REVIEW

REVIEWER	Samantha Meints, Ph.D. Brigham and Women's Hospital, Harvard Medical School, United States
REVIEW RETURNED	18-Nov-2019

GENERAL COMMENTS	Per reviewer #2's comment, the authors still did not summarize the past literature regarding the relationship between CS and disability. Instead, they added a couple references to show there is a scarcity of work. This does not sufficiently address the reviewer concerns. Otherwise, the authors did a nice job addressing reviewer concerns.
---

VERSION 3 – AUTHOR RESPONSE

Reviewer #1, dr. Samantha Meints, comments:

Per reviewer #2's comment, the authors still did not summarize the past literature regarding the relationship between CS and disability. Instead, they added a couple references to show there is a scarcity of work. This does not sufficiently address the reviewer concerns.

Otherwise, the authors did a nice job addressing reviewer concerns.

Response: Our sincere apologies for not adequately dealing with the issue. We unintentionally misinterpreted reviewer's comment, resulting in a limited and not adequate reaction. Specifically, our search on the association between CS and functioning was restricted to the instruments chosen for our study instead of the constructs themselves. We then were unable to find relevant publications as to add a summary of it. We have expanded on the search and made a summary of the scarce literature available. We expect to have now appropriately addressed reviewers concerns.

The following addition can be found in the manuscript (page 5 lines 18-24):

“More severe symptomatology of CS, decreased parasympathetic/vagal activity, or altered somatosensory responses have been associated with greater self-reported pain-related disability.[Kregel et al. 2018, Huysmans et al. 2018, Tanaka et al. 2019, Cohen et al. 2000, Gockel et al. 2008, Hübscher et al. 2013, Georgopoulos et al. 2019] However, the associations between altered somatosensory responses and self-reported disability, were dependent on the site of testing, the stimuli, and the type of tests performed. Also, CS has been assessed with different instruments that measure specific CS clinical symptoms,[den Boer et al. 2019] lacking a comprehensive evaluation of the mechanisms involved in CS. Moreover, functioning has most frequently been assessed with self-reported disability which is not exclusively representative of the functioning status of the patient.”

References:

Kregel J, Schumacher C, Dolphens M, et al. Convergent validity of the dutch central sensitization inventory: associations with psychophysical pain measures, quality of life, disability, and pain cognitions in patients with chronic spinal pain. Pain Practice. 2018;18(6):777-87.

Huysmans E, Ickmans K, Van Dyck D, et al. Association Between Symptoms of Central Sensitization and Cognitive Behavioral Factors in People With Chronic Nonspecific Low Back Pain: A Cross-sectional Study. J Manipulative Physiol Ther. 2018;41(2):92-101.

Tanaka K, Murata S, Nishigami T, et al. The Central Sensitization Inventory predict pain-related disability for musculoskeletal disorders in the primary care setting. Eur J Pain. 2019; 23(9):1640-8.

Cohen H, Neumann L, Shore M, et al. Buskila. Autonomic dysfunction in patients with fibromyalgia: application of power spectral analysis of heart rate variability. Semin Arthritis Rheum. 2000;29(4):217-27.

Gockel M, Lindholm H, Niemistö L, et al. Perceived disability but not pain is connected with autonomic nervous function among patients with chronic low back pain. J Rehabil Med. 2008;40(5):355-8.

Hübscher M, Moloney N, Leaver A, et al. Relationship between quantitative sensory testing and pain or disability in people with spinal pain—A systematic review and meta-analysis. Pain. 2013; 154(9), 1497–504.

Georgopoulos V, Akin-Akinyosoye K, Zhang W, et al. Quantitative sensory testing and predicting outcomes for musculoskeletal pain, disability, and negative affect. Pain. 2019;160(9):1920-32.

den Boer C, Dries L, Terluin B, et al. Central sensitization in chronic pain and medically unexplained symptom research: A systematic review of definitions, operationalizations and measurement instruments. *J Psychosom Res.* 2019;117:32–40.

VERSION 4 – REVIEW

REVIEWER	Samantha Meints Brigham and Women's Hospital, USA
REVIEW RETURNED	20-Dec-2019

GENERAL COMMENTS	 • Augmented pain processing does not always mean central sensitization. Indeed, it could be peripheral sensitization. This should be addressed • The abstract should be written in complete sentences. • On page 4, line 25, authors indicate that the described instruments “comprehensively” measure indicators of CS. This is quite an overstatement. • On page 5, line 5, authors indicate that a subsample of CLBP patients have CS. This needs a citation. • On page 5, line 12, the authors are discussing CS symptomology and then switch to mechanisms. Just because an instrument highlights a CS symptom does not allow for identification of mechanisms. This should be tempered. • Based on the introduction, it remains unclear why the instruments assessing CS as well as those measuring functioning were chosen. This would help provide a clear rationale for the indicated study. • Overall, the inclusion and exclusion criteria should be more detailed. It is unclear how these were assessed (e.g., competence, comorbidity, etc.). Moreover, there should be specific disorders listed rather than just a few examples. What about opioid use or other pain-related medication use that can impact CS, pain, and function? • It is unclear how much of the measures are part of the standard of care and how much is added. This should be clarified. • What is done for WUR if NRS-1 = 0? • Citations needed for expected medium correlations between CS measurements. Also, because there are multiple correlation analyses, there is a higher risk for Type I error. How will this be addressed?
---

VERSION 4 – AUTHOR RESPONSE

Reviewer #1, dr. Samantha Meints, comments:

1. Augmented pain processing does not always mean central sensitization. Indeed, it could be peripheral sensitization. This should be addressed

Response: Thank you for the remark, we have now specified that it is augmented central pain processing:

“A relevant subsample of patients with Chronic Low Back Pain (CLBP) have manifested augmented central pain processing, Central Sensitisation (CS).”

Changes to the manuscript: page 2 line 3.

2. The abstract should be written in complete sentences.

Changes to the manuscript: page 2 lines 11-15.

3. On page 4, line 25, authors indicate that the described instruments “comprehensively” measure indicators of CS. This is quite an overstatement.

Response: We agree with the reviewer and, consequently, have deleted the concerning sentence.

4. On page 5, line 5, authors indicate that a subsample of CLBP patients have CS. This needs a citation.

Response: We added the following reference:

Giesecke T, Gracely RH, Grant MAB, et al. Evidence of augmented central pain processing in idiopathic chronic low back pain. *Arthritis Rheum.* 2004;50(2):613–23.

5. On page 5, line 12, the authors are discussing CS symptomology and then switch to mechanisms. Just because an instrument highlights a CS symptom does not allow for identification of mechanisms. This should be tempered.

Response: Thank you for pointing this out to us. We, indeed, intended to focus on clinical symptoms, not on mechanisms. We have corrected the text accordingly:

“Also, CS has been assessed with different instruments that measure specific CS clinical symptoms, lacking a comprehensive assessment of CS in which various reference clinical symptoms are measured together.”

Changes to the manuscript: page 5 lines 12-14.

6. Based on the introduction, it remains unclear why the instruments assessing CS as well as those measuring functioning were chosen. This would help provide a clear rationale for the indicated study.

Response: Following your suggestion, we have reorganized information from CS and functioning paragraphs to further develop the last paragraph of the introduction, in order to illustrate better the link between CS and functioning with their operationalization instruments.

Changes to the manuscript: page 4 lines 22-25; page 4 line 31 to page 5 line 4; and page 5 lines 20-28.

7. Overall, the inclusion and exclusion criteria should be more detailed. It is unclear how these were assessed (e.g., competence, comorbidity, etc.). Moreover, there should be specific disorders listed rather than just a few examples. What about opioid use or other pain-related medication use that can impact CS, pain, and function?

Response: The eligibility of patients to participate is decided based on their file and during the first visit with the rehabilitation physician. For clarification, we have repeated in the procedure section that the eligibility criteria are based on patient’s file during the first appointment with the rehabilitation physician.

Regarding the use of medication, the use of opioid or pain medication is not a criterion for exclusion, and the information is collected. The information about the use of medication may enter the multiple correlation analyses and be selected as confounder for the multiple regression analyses; depending on their partial correlation coefficient with functioning outcomes and provided it fulfills with the $p \leq 0.10$ requirement (see Methods Section, page 18 line 6-7).

We have described the inclusion and exclusion criteria for participants in more detail in the protocol. It

includes a definition of chronic primary low back pain condition within the ICD-11 framework [IASP taskforce 2019], it specifies mental and language competences, and the disorders susceptible for exclusion and the CS-related disorders for inclusion, as well as the use of medication.

Changes to the manuscript: page 6 line 27 to page 7 line 15; and page 7 line 22.

Reference:

IASP Taskforce for the Classification of Chronic Pain. The IASP classification of chronic pain for ICD-11: chronic primary pain. *Pain*. 2019; 160(1): 28-37.

8. It is unclear how much of the measures are part of the standard of care and how much is added. This should be clarified.

Response: Thank you for the comment. We described more clearly the procedures with regards to the distinction between care as usual and additional measurements as well as in the measurement points.

Changes to the manuscript: page 7 line 28 to page 8 line 5.

9. What is done for WUR if $NRS-1 = 0$?

Response: The WUR calculation as described in the original DFNS protocol [Rolke et al. 2006] will be performed: $WUR = (NRS-10 / NRS-1)$. If a patient reports WUR $NRS-1=0$ at any of the body locations, the WUR for that location cannot be calculated (due to the denominator being 0) and it will be registered as a missing value.

Because the same might occur with DMA $NRS-1$, we have changed this accordingly.

Changes to the manuscript: page 9 line 20 (DMA); and page 10 line 7 (WUR).

Reference:

Rolke R, Baron R, Maier C, et al. Quantitative sensory testing in the German Research Network on Neuropathic Pain (DFNS): Standardized protocol and reference values. *Pain*. 2006;123(3):231–43.

10. Citations needed for expected medium correlations between CS measurements. Also, because there are multiple correlation analyses, there is a higher risk for Type I error. How will this be addressed?

Response: We added the following reference to the correlations between CS measurements: Kregel J, Schumacher C, Dolphens M, et al. Convergent validity of the dutch central sensitization inventory: associations with psychophysical pain measures, quality of life, disability, and pain cognitions in patients with chronic spinal pain. *Pain Practice*. 2018;18(6):777-87.

Regarding the type I error, we are aware of the risk of it due to the multiple correlation analyses. In response 1 to the reviewers, we decided to have a stricter significance level, i.e. 0.01 (see also answer of R1- comment 14, copied below). We have it specified in the protocol as well.

Changes to the manuscript: page 17 line 21 and page 18 lines 9-12, and page 21 line 10.

“Reviewer: There are a few aspects of the statistical analyses that require attention. Firstly, it is proposed that multiple correlations are to be performed. However, there is no mention of the possible issues with this and how they might be overcome.

Response: There are a large amount of multiple correlation analyses to be performed in order to respond the main objectives. We are aware that there is a great risk of type I error and the idea of performing Bonferroni adjustments has been considered, but:

1) Due to the large amount of correlations in the multiple correlation analyses, a Bonferroni adjustment would lead to an increase in Type II error when trying to avoid Type I error.

2) Bonferroni would only adjust for the significance. Establishing results and conclusions only on significance can be flawed [Ziliak and McCloskey 2008]; moreover, we are interested in the size of effect of the associations which would not change even if the significance is adjusted.

In this situation and all things considered to have a stricter significance level for multiple correlation analyses, i.e. 0.01; and to provide 95% Confidence Intervals of the correlation coefficients in the results because they can be more insightful of the values of the population. We believe that with these measures we will be able to extract meaningful information.

Reference:

Ziliak S, McCloskey DN. The cult of statistical significance: How the standard error costs us jobs, justice, and lives. University of Michigan Press; 2008.”

Additional changes to the protocol:

1. During our revision in the original DFNS protocol and reference values by Rolke et al.(2006), we have come to realize that it would not be possible to compare our measurements to theirs—DFNS measurements include the face, hands and feet, whereas ours include the upper back, low back, and upper legs. As a result z-scores for our study based on their reference values will not be possible. We have decided to present an overall mean for each QST test. We believe that with this strategy we'll be able to provide informative results about the general somatosensory profile of patients with CLBP. The aforementioned WUR (Question n.9) might not be the only QST test with some missing values. We have resolved this by calculating the overall mean for each QST test if at least four out of six locations have data. With this method we intend to obtain overall means that are representative of the sample even if there could be some power loss, although it is expected to be minimal.

Changes to the manuscript: Page 16 lines 23-25.

Reference:

Rolke R, Baron R, Maier C, et al. Quantitative sensory testing in the German Research Network on Neuropathic Pain (DFNS): Standardized protocol and reference values. Pain. 2006;123(3):231–43.

2. The QST assesses the somatosensory system with a battery of different sensory tests. It might be possible that the different QST tests relate differently to the functioning domains. The pre-selection of a single test representative of the whole battery of QST tests that could relate to all functioning domains, might be too limited and much information may be missed. To overcome this, the selection of the QST test to enter the multiple regression models for RQ2 and RQ3, will be based on the highest correlation coefficient per functioning dependent variable.

Changes to the manuscript: Page 17 line 2 and page 21 line 11.